# Tools for mapping multi-scale settlement patterns of building footprints: An introduction to the R package *foot*

**Warren C. Jochem** **\*, Andrew J. Tatem**

WorldPop, School of Geography and Environmental Science, University of Southampton, Southampton, United Kingdom

\* w.c.jochem@soton.ac.uk

**Data Availability Statement:** All data supporting this study are openly available from the University of Southampton repository at https://doi.org/10.5258/SOTON/D1674.

## Abstract

Spatial datasets of building footprint polygons are becoming more widely available and accessible for many areas in the world. These datasets are important inputs for a range of different analyses, such as understanding the development of cities, identifying areas at risk of disasters, and mapping the distribution of populations. The growth of high spatial resolution imagery and computing power is enabling automated procedures to extract and map building footprints for whole countries. These advances are enabling coverage of building footprint datasets for low and middle income countries which might lack other data on urban land uses. While spatially detailed, many building footprints lack information on structure type, local zoning, or land use, limiting their application. However, morphology metrics can be used to describe characteristics of size, shape, spacing, orientation and patterns of the structures and extract additional information which can be correlated with different structure and settlement types or neighbourhoods. We introduce the *foot* package, a new set of open-source tools in a flexible R package for calculating morphology metrics for building footprints and summarising them in different spatial scales and spatial representations. In particular our tools can create gridded (or raster) representations of morphology summary metrics which have not been widely supported previously. We demonstrate the tools by creating gridded morphology metrics from all building footprints in England, Scotland and Wales, and then use those layers in an unsupervised cluster analysis to derive a pattern-based settlement typology. We compare our mapped settlement types with two existing settlement classifications. The results suggest that building patterns can help distinguish different urban and rural types. However, intra-urban differences were not well-predicted by building morphology alone. More broadly, though, this case study demonstrates the potential of mapping settlement patterns in the absence of a housing census or other urban planning data.

## Introduction

Accurate and complete maps of buildings are a foundational data layer for researchers and practitioners seeking to understanding cities and characteristics of the built environment. Identifying and mapping the footprints of structures and their agglomerations into human settlements is a first step towards improving our understanding of local population patterns,

**Funding:** Funding support comes from the Bill and Melinda Gates Foundation and the United Kingdom Foreign, Commonwealth & Development Office as part of the Geo-Referenced Infrastructure and Demographic Data for Development project (GRID3) (OPP1182425). Project partners in GRID3 include the WorldPop Research Group, the United Nations Population Fund, the Flowminder Foundation, and the Center for International Earth Science Information Network within the Earth Institute at Columbia University.

**Competing interests:** The authors have declared that no competing interests exist.

providing services and utilities to all areas, and mapping building stock and urban extents [1–4]. Within the field of planning and particularly the area of urban morphology, the form and the patterns of buildings in space have been a means to explore the history of cities and the political economic and social forces shaping their developments [5, 6]. The visible patterns and features of the built environment can be quantified with a wide range of morphology metrics [7]. These metrics describe characteristics (e.g. size, shape, density, compactness, arrangement) of individual buildings or within areas, which can be used to describe urban context at a fine spatial scale [8].

Building footprint datasets are also becoming key inputs for research in geographic information science and other related areas. For example, building footprints have been used to identify areas at risk of the 100 year floodplain [9]. Footprints have also been used as part of building models to estimate structure age for energy consumption [10] and to model rooftop solar potential [4]. Other recent analyses involving building datasets have included delineating urban areas based on building densities for all of France [11] and Spain [1] and deriving settlement types [12]. These applications have not always explicitly engaged with past research on urban morphology and urban planning; however, they often share similarities in their approaches of using or extracting information based on building footprint datasets. Specifically, within much of the work involving building maps there is often a goal to identify or classify similar patterns of morphometric characteristics. The identified classes can then help distinguish intra-urban neighbourhoods, differentiate settlement types or periods of development [3, 8, 13, 14].

Across many research areas there is a trend towards big data analysis to explore urban areas [5], including using building datasets and morphology metrics. This trend has been noted previously as part of the growing use of computational methods and larger datasets more generally and particularly in geographic data science [15]. The opportunities for these types of analyses are increasing as spatial databases of building footprint polygons with complete and consistent coverage for large regions and entire countries are becoming more readily available. Some governments and national mapping agencies already make such geospatial building datasets openly available. In Great Britain there is the Ordnance Survey OpenMap Local [16]. Some major cities including New York City [17], Chicago [18], and Washington, DC [19] provide their own data, among other examples. Finally, volunteered geographic information (VGI), such as OpenStreetMap (www.openstreetmap.org), is another source, producing building coverage through manual digitising from imagery or by incorporating open building datasets.

The growth of very high spatial resolution imagery (sub-metre) and improvements in computational power and algorithms are enabling another source of building footprints from automatic extraction and mapping from overhead imagery. Recent research has explored the potential for using deep learning techniques such as neural networks to produce pixel-level labelling of buildings [20, 21]. The growth of computing resources is enabling such automated building extraction algorithms to be scaled up to cover whole countries. For example, Microsoft used a convolutional neural network to extract 125 million building footprints from imagery across the United States. A post-extraction processing step was applied to create more regularly shaped polygons. They later applied a similar method to produce building footprints across Uganda and Tanzania [22, 23]. The Microsoft building footprint datasets are openly available (https://github.com/Microsoft/USBuildingFootprints). Facebook Analytic Labs also implemented building feature extraction from high resolution imagery for 140 countries [24]; however, the only publicly available data from this work have been aggregated to a 1 arc-second (approximately 30 m) resolution. The remote sensing-derived building polygon datasets complement other spatial vector databases of building footprints and are increasing the

availability and geographic coverage of such building data, particularly in resource-poor settings or low-/middle-income countries which may lack other sources of information on urban planning or the built environment.

With the growth in available data for building footprints from a range of sources, researchers and practitioners need software tools to enable effective use of the larger data sources and to extract the variety of metrics suitable for further analysis and interpretation. The purpose of this paper is to introduce the *foot* package [25], a set of open-source software tools as part of the R computing language [26] for calculating common building footprint metrics. These tools are provided to support flexible workflows applied to 2D spatial polygon building representations and to calculate morphology metrics suitable for many different applications. We demonstrate the use of the *foot* tools to calculate building pattern metrics for all of England, Scotland, and Wales. We use the resulting gridded summary metrics of building patterns to develop a simple settlement classification. This example is intended to demonstrate how new information can be extracted from building footprint datasets. This approach may be of more value in low- and middle-income countries where up-to-date data on land use or neighbourhood types do not already exist, but where new building footprint datasets are becoming available. All results from this study are openly available from the University of Southampton repository at https://doi.org/10.5258/SOTON/D1674.

## Calculating building footprint metrics

The R package *foot* consists of a set of functions to calculate common morphology measures of vector representations of building footprints. These functions can be applied at the scale of the individual footprint shape, but the package also contains functions to summarise the measures for differing levels of geography. The main calculations are combined into a function to implement common workflows. The *foot* package is coded entirely in the R statistical computing language, making use of the *sf*, *stars*, and *lwgeom* packages [27–29] to access external spatial data libraries.

In the following sections we detail the basic use of the package and some of its key features. Development of the package is ongoing and the latest version of the source code is available under a GPL-3.0 open source license from Github (https://www.github.com/wpgp/foot).

### Basic use and available metrics

After installing the *foot* package and its dependencies (see S1 Text), the package can be loaded in an R session. A small sample of building footprints are provided with the package which were publicly released and licensed by Microsoft under the Open Data commons Open Database License (ODbL v1.0). These data are used for the demonstrations in this section. Further details of the package are available from the documentation (see? `foot`) and in three tutorial vignettes installed with the package and available in the supplementary materials (S2 Text).

The available metrics calculate area, perimeter, roundness, compactness, angle of rotation, and nearest neighbour distance. These measures can be summarised for a user-defined region with a total, count, mean, median, min, max, standard deviation, coefficient of variation, nearest neighbour index, or entropy. In addition, there are options for a binary indicator of footprint present (or not) and a count of footprints per zone. Not all summaries are available for all measures, as appropriate. The list of measures is given in Table 1, and the names of the functions to calculate the metrics can be listed in an R session using `list_fs()`.

The shape index implemented is calculated as the ratio of footprint area to the area of the minimum bounding circle which contains the footprint polygon. The shape values can range from 0 to 1, indicating more complicated shapes to less complicated or more circular shapes.

**Table 1. Currently available measurements and summary statistics available in the *foot* package.**

| | Area | Perimeter | Rotation Angle | Nearest Neighbour Distance | Shape | Compactness |
|---|---|---|---|---|---|---|
| Mean | X | X | X | X | X | X |
| Median | X | X | X | X | X | X |
| Standard Deviation | X | X | X | X | X | X |
| Min, Max | X | X | X | X | X | X |
| Coefficient of Variation | X | X | X | X | X | X |
| Total (sum) | X | X | | | | |
| Entropy | | | X | | | |
| Nearest Neighbour Index | | | | X | | |

Summary statistics are available for a range of characteristics (area, perimeter, angle of rotation, nearest neighbour distance, shape, compactness) for building footprint polygons.

The compactness measure implemented is the Polsby-Popper index [30], which is calculated from the area (*a*) and perimeter (*p*) for any building footprint *i* as: $c_i = \frac{4\pi a_i}{p_i^2}$. A value of 1 indicates the most compact shape while values approaching zero reflect shapes with no compactness. The metrics are introduced in more detail in the package documentation and vignettes. These metrics were chosen as commonly used morphology measures to quantify the dimension, shape, and distribution of building features in local areas [7]. Future developments in *foot* will allow users to define their own metrics and summary functions.

Some less-conventional measures are also implemented in the *foot* package. The angle is a measure of structure orientation. It is calculated as the heading, in degrees, of the rotated minimum bounding rectangle. To summarise the orientations for a local area, the angle (along with its 180 degree opposite) are binned into 10-degree categories and the Shannon entropy value (*H*) is calculated [31]. For each zone, entropy is calculated as: $H = -\sum_{i=1}^{n} P(o_i) \log_e P(o_i)$, where $P(o_i)$ represents the proportion of building orientations in each bin, *i*, out of *n* total bins. Additionally, this entropy can be normalised (the default setting in *foot*) to describe the local deviation from a hypothetical perfect grid of structures, potentially suggesting areas with more (or less) formal planning and similarly oriented structures. The angle entropy measure is based on a study of street network orientations demonstrated by Boeing [32]. The nearest neighbour index (NNI) is more commonly used in spatial point pattern analysis [33], but it can be used to quantify patterns of building centroid points and summarise the tendency to spatially cluster or be dispersed within a geographic region. NNI is based on comparing the observed average nearest neighbour distance to a hypothetical distance that could be expected if points were randomly distributed in the same area. NNI is calculated as: $NNI = \frac{\frac{\sum_{i=1}^{n} d_i}{n}}{.5\sqrt{\frac{A}{n}}}$, where *d* is the nearest neighbour distance for building *i*, out *n* total buildings, and *A* is the total area of the zone being evaluated. This measure has been applied previously to detect differences in residential areas [34].

An example of basic usage is shown in Code Block 1, below. The core functionality is accessed through `calculate_footstats`. All functions in the *foot* package follow the same naming template. The first argument to a function is always a footprint dataset which allows for piping syntax using the %>% operator, if the user prefers. The characteristics to calculate are specified in the "what" argument. Users can specify the units of measure for the morphology characteristics. The default return value for summary functions is a data table [35] with measurement units [36] or a Geotiff for gridded outputs.

**Code block 1. Basic usage of the *foot* package.** The code snippet demonstrates how the package can be used to calculate building-level and summary measures of basic area and perimeter characteristics. The default return value for all foot functions is a data table. Functions shown are applied to a sample dataset provided with the R package code.

```
# load package
library(foot)
# list available metrics
list_fs()
# load sample data
data("kampala", package = "foot")
buildings <- kampala$buildings
# calculate building-level measures
calculate_footstats(buildings, what = c("area", "perimeter"))
# calculate average area of all footprints
calculate_footstats(buildings,
  what = "area",
  how = "mean")
# calculate the coefficient of variation for all perimeters
calculate_footstats(buildings,
  what = "perimeter",
  how = "cv")
```

The function `calculate_footstats` is a convenience function to also support calculating multiple metrics and creating area-level summary measures of footprint characteristics. An example is demonstrated in Code Block 2. To create area-level summary measures a user must first define the group to which a building belongs. Most commonly this grouping will be a geographically-defined zone (e.g. an administrative unit), but in theory any classification assigned at the building level could be used as a grouping variable for these analyses. Summary calculations also require a valid summary statistic (Table 1) for the characteristic, specified as one or more parameters to the "how" argument.

**Code block 2. Defining geographic zones and calculating area-level summaries.** The `calculate_footstats` function supports area-level summary measures of footprint morphology characteristics by allowing users to supply spatial datasets of footprint polygons and area zones. Multiple summary metrics can be calculated on the footprints and the result is a table of summary measures.

```
# load sample polygon zones from package data
adminzones <- kampala$adminZones
# Creates zonal index and calculates multiple metrics
results <- calculate_footstats(buildings, # supply footprints
     zone = adminzones, # supply zonal polygons
     what = c("area", "perimeter"),
     how = c("mean", "cv"))
  print(results)
# alternatively, buildings within zones can be pre-defined
bldgsZone <- zonalIndex(buildings,
    zone = adminzones,
    method = "centroid", # or 'intersect' or 'clip'
    returnObject = TRUE)
# new set of building footprints with `zoneID`added
  bldgsZone
# calculate multiple metrics with pre-defined zone identifiers
results2 <- calculate_footstats(bldgsZone,
     zone = "zoneID", # supply column name
     what = c("area", "perimeter"),
     how = c("mean", "cv"))
```

```
print(results2)
```

Identifying the buildings' geographic zone can be accomplished as a pre-processing step in a GIS, with other spatial tools in R, or by using functions provided in the foot package. The `zonalIndex` function spatially links footprints and geographic zones and provides a unique ID for calculations. This function is used internally by `calculate_footstats` when a user supplies a spatial polygon as the argument to the zones.

The zonal index function has further parameters for greater control when defining inclusion. A user can define the buildings belonging to a zone in three ways: 1) as any polygons whose centroid point intersects a zone; 2) as any polygons which intersect a zone, or, 3) building footprint shapes can first be clipped to the bounds of the zone and then intersected (Fig 1). Note that when intersecting building polygons with zones, buildings which overlap multiple zones will be included in each zone, duplicating the building record. This side-effect can be desirable for smoothing style analyses with overlapping moving windows. Additionally,

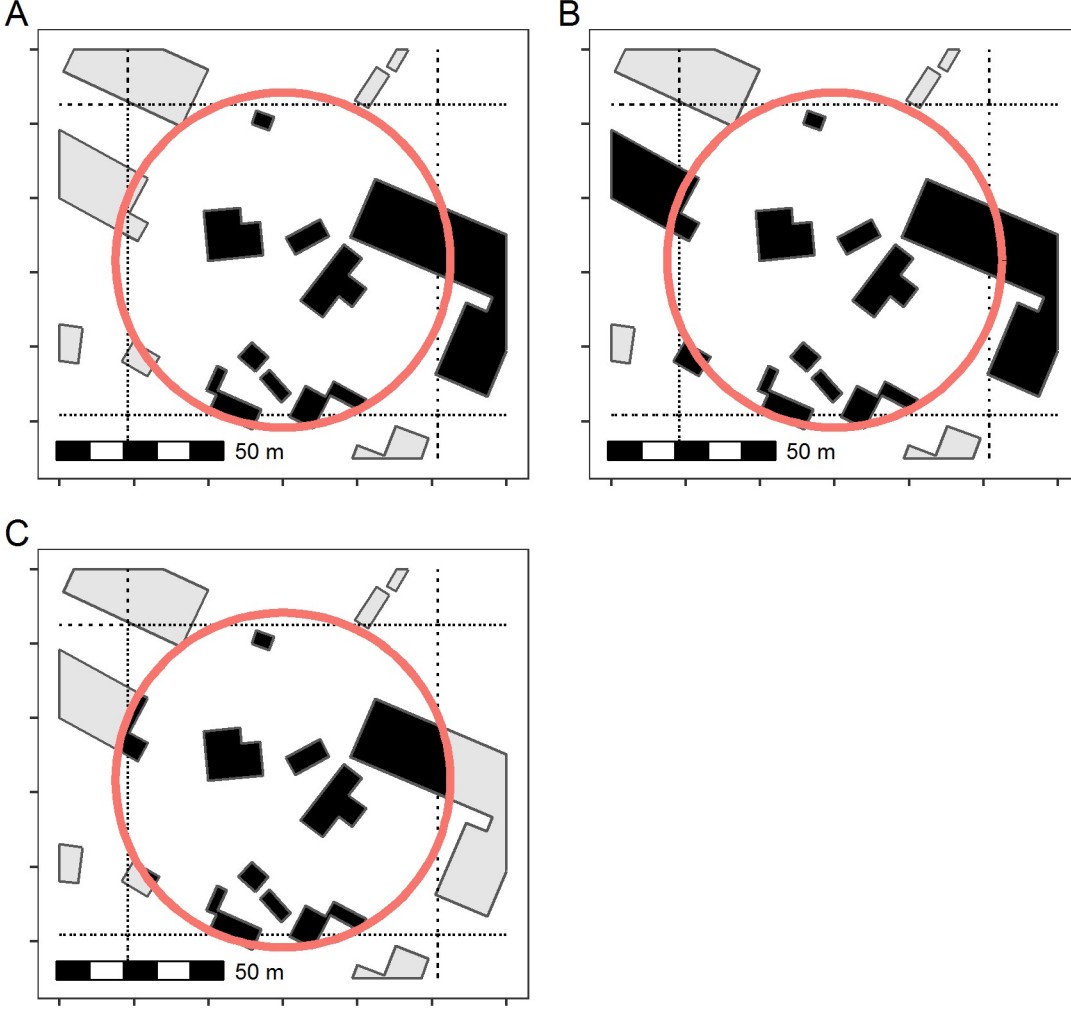

**Fig 1. Defining building footprints within a geographic area using the zonalIndex function.** Users can provide shapes to define a zone. Shaded regions of polygon footprint are included in the morphology calculations and summary measures for the red "zone." The building footprint shapes can be included in the zone if their centroids intersect the zone (A), if any part of the footprint intersects the zone (B), or the footprint shapes can be clipped to the zone bounds (C). Building footprint data shown in this figure are made available by Microsoft under the Open Data Commons Open Database License (ODbL v1.0).

clipping building shapes could introduce small slivers, but these polygons can be filtered out in the `calculate_footstats`. The footprint clipping approach may be necessary for certain types of analyses if, for instance, exact building areas within a plot are desired. By default, however, `zonalIndex` uses the centroid point of the building polygons to intersect with the zones. This process is faster, prevents duplicating features (unless zones overlap), and avoids slivers from clipping.

## Multiple representations

One of the strengths of the *foot* package is its flexible design to allow output summaries for multiple scales and geographic representations. These scales include the building level, areal level (as demonstrated in Code Blocks 1 and 2), and for a gridded dataset as shown in Fig 2. Output at the building-level is the most granular level where each building polygon has its geometric features calculated (Fig 2A). This level of output might be used to help characterise individual building use or residential vs. non-residential function [37, 38]. The areal level provides zonal summaries of metrics for units and allows users to characterise differences among administrative units, city blocks, or other areas (Fig 2B). The zones do not have to be extensive lattices, they can be separate survey plots for example. As noted above, how to define which (parts of) buildings are included within a zone allows for different representations and flexibility in analyses. The concept of a zonal summary measure can be extended to regularly shaped grid (Fig 2C and 2D), discussed further in the next section.

## Grids and spatial resolutions

Calculating summary measures of building morphology on a regular grid, or spatial raster, provides certain advantages. From a data management perspective, grids are often easier to store and manipulate than geodatabases containing millions of footprints [39]. But there are also advantages from an analytical perspective. Morphology calculations have often been tied to the scale of a "plot" defined by property lines, roads or Thiessen polygons [8, 40]. However, this decision could be problematic in places without pre-defined small urban zones or lacking good data on roads to define these areas. A gridded morphology dataset can also begin to provide a landscape perspective of the variations in the built environment [12]. Grids allow for the modifiable area unit problem (MAUP) to be examined through easily changed origins and resolutions. Moreover, a gridded format for building morphology can allow these metrics to be more easily integrated with other gridded geospatial data layers [e.g. 41] to support additional spatial modelling and analyses.

The *foot* package provides a second main function specifically designed to create gridded summary statistics of building morphology metrics, such as shown in Fig 2C and 2D. The `calculate_bigfoot` function takes footprint shapes and a template raster defining the extent and resolution as its inputs. Similar parameters are specified for the "what" and "how" arguments for characteristics and summary statistics, respectively. The function is designed with computational efficiency in mind for producing country-scale datasets, as demonstrated in the case study presented in the next section. Internally, the function creates bounding box queries to extract and process only small subsets of the data. These processing steps can be done in parallel on multiple processing cores when sufficient memory is available. The grid cells of the template raster serve as the "zones" for summary calculations, and similar to the `calculate_footstats` function, can allowing for shape clipping in the inclusion criteria.

This function also introduces a parameter for a focal radius to calculate metrics within a wider area than a single grid cell. The focal radius establishes a circular processing window centred on each grid cell of the template raster. Varying the focal radius provides further

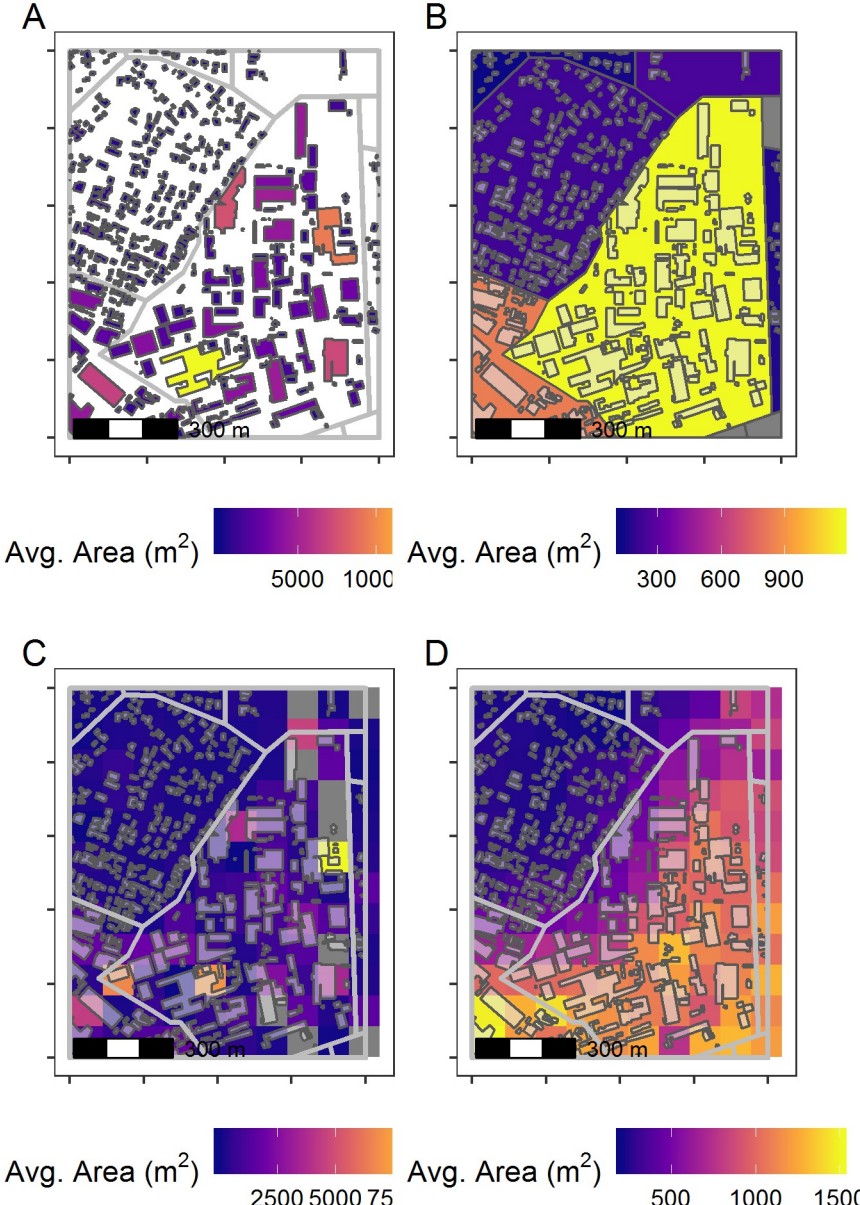

**Fig 2. Morphology metrics summarised in different representations.** The foot package calculations can include the building level (A), the area level (B), or on regular grids without (C) or with (D) overlapping local windows to create a smoothed summary calculation. Building footprint polygons are overlaid on Fig 2B, 2C, and 2D. Building footprint data shown in this figure are made available by Microsoft under the Open Data Commons Open Database License (ODbL v1.0).

flexibility when processing gridded data to represent the building patterns within different spatial scales which can help to describe the local contexts [12, 34]. Example calculations of gridded outputs are shown in Fig 3 using the code in Code Block 3. Note that with a high spatial resolution template grid and a large focal radius, the windows from neighbouring grid cells will overlap. This is a potentially desirable effect and creates a smoothed summary measure, though building footprints are effectively counted multiple times into each overlapping zone. The choice of spatial resolution and whether to use a focal window depends on the specific application, but the options are available in the *foot* package.

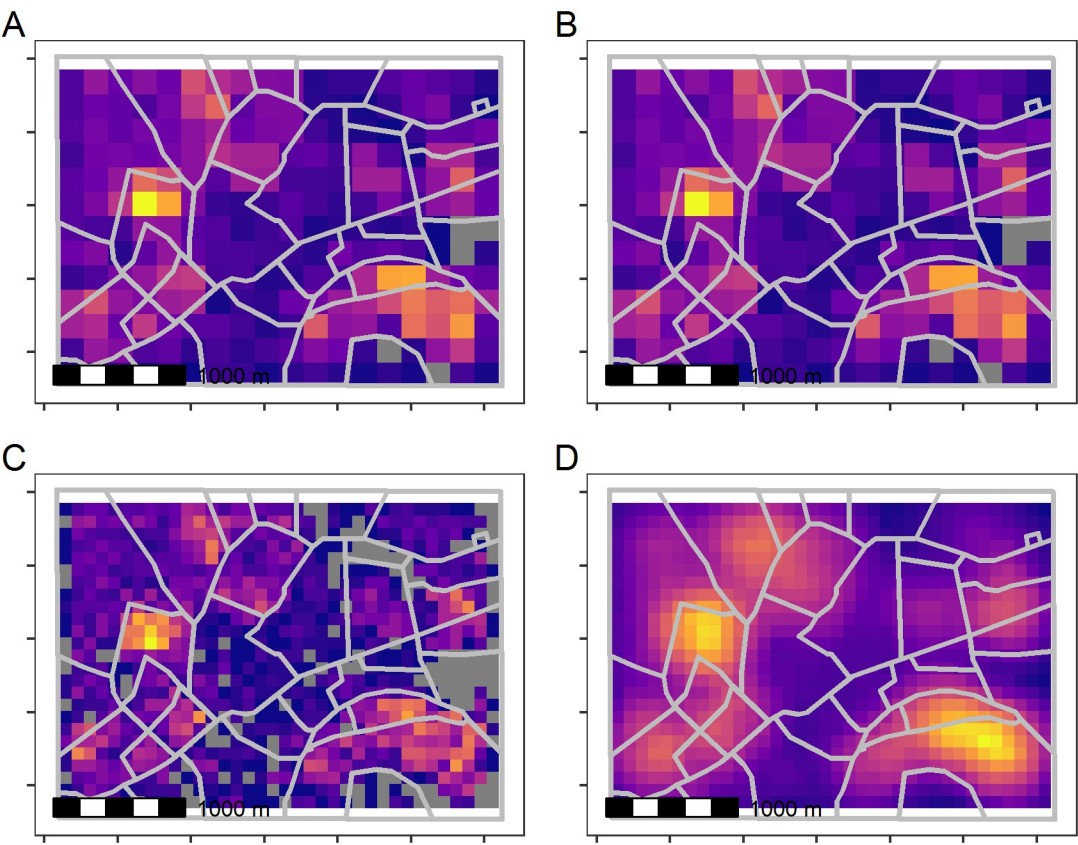

**Fig 3. Varying resolution and focal radius in gridded summaries of building counts.** The output spatial resolution can be varied (A and B) and this can be used in conjunction with a circular window with a user-defined focal radius (C and D) to produce gridded summaries. Data shown are the authors' calculations using building footprints are made available by Microsoft under the Open Data Commons Open Database License (ODbL v1.0).

**Code block 3. Calculating gridded representations of footprint morphology metrics.**
The `calculate_bigfoot` function allows for summaries to be calculated within grid cells defined by a template raster, or within user-defined circular moving windows. The results of this sample code are shown graphically in Fig 3.

```
# load building footprints
data("kampala", package = "foot")
buildings <- kampala$buildings
# load template grid with 100m resolution
g <- kampala$mastergrid
# change resolution
g50 <- raster::disaggregate(g, fact = 2)
g200 <- raster::aggregate(g, fact = 2)
# varying template grid resolution
# Figure A
k50 <- calculate_bigfoot(buildings,
  what = "settled",
  how = "count",
  template = g50,
  parallel = FALSE,
  verbose = TRUE)
```

```
# Figure B
k200 <- calculate_bigfoot(buildings,
  what = "settled",
  how = "count",
  template = g200,
  parallel = FALSE,
  verbose = TRUE)
# varying focal radius of moving window
# note: template grid resolution remains fixed
# Figure C
r50 <- calculate_bigfoot(buildings,
  what = "settled",
  how = "count",
  template = g,
  focalRadius = 50,
  parallel = FALSE,
  verbose = TRUE)
# Figure D
r300 <- calculate_bigfoot(buildings,
  what = "settled",
  how = "count",
  template = g,
  focalRadius = 300,
  parallel = FALSE,
  verbose = TRUE)
```

## Case study: Building patterns in Great Britain

We demonstrate the use of the *foot* package as well as the applicability of gridded output representations through a case study of the building patterns in Great Britain. The data used are 2D building footprints from Ordnance Survey's OS OpenMap Local for 2018 (Contains Ordnance Survey data © Crown copyright and database right 2018). These data were publicly released under the Open Government License (OGL v3.0). The data were retrieved as a single, merged GeoPackage file [42]. The dataset contains 2D building footprints as polygons for all of England, Scotland, and Wales. In order to produce a gridded output, we first created a 100 m x 100 m resolution grid covering the entire landmass defined by Ordnance Survey. This grid serves as the template for the extent and resolution of the output metrics. The resolution for the grid was chosen from initial test runs as a compromise to reduce output file size while maintaining sufficient resolution to detect local changes in building patterns. The *foot* package (version 0.6) and the `calculate_bigfoot` function was then used to calculate all metrics. Two sets of morphology summary metrics were calculated. In the first set, no focal window was used and buildings were summarised into intersecting 100 m grid cells without clipping. In the second set of outputs, a 250 m focal radius window centred on each template grid cell was added and again buildings were summarised into all intersecting areas without clipping.

Using the gridded building pattern layers from the first step, we proceeded to create a settlement typology map by clustering and grouping the grid cells based on the morphology values. We applied a Gaussian mixture modelling approach using the R package *mclust* [43, 44]. Gaussian mixture models (GMMs) are a model-based clustering algorithm which use multivariate normal distributions to describe any grouping in the data. The number and size of these distributions are treated as unknowns in the model which are then fit using expectation-maximisation (EM), leading to an unsupervised clustering of the observed data points. GMMs have more flexibility than other unsupervised methods, such as K-means, because they can allow for varying volume, shape, and orientation of the clusters in data space. In order to select

the best performing clustering model, we fit mixture models with between 2 and 15 potential groups while allowing fully varying covariance structure (volume, shape, and orientation). A Bayesian information criterion (BIC) score is used to compare the models and select the best fitting number of groups [44]. Using the selected number of groups, we predicted the settlement type for each 100 m grid cell based on the maximum predicted probability of group assignment. We also applied a 3 x 3 cell majority filter to smooth the predictions. This method of using GMMs for clustering has been used previously for settlement classification mapping [12]. We chose an unsupervised, model-based method as this is an exploratory analysis meant to be examined and compared with other urban delineations and classifications; however, alternative, supervised clustering methods could be used with the morphology metrics to improve on a settlement typology.

The resulting layers of footprint metrics and the settlement type map were examined visually and then compared them with two existing settlement maps for England and Wales by summarising the majority settlement type to the Census Output Area (OA). We compare our results first to the rural-urban classification (RUC) for 2011 OAs [45]. The RUC classification is based on physical settlement form and the density of residential dwelling locations. Rural and urban areas are further divided into broad categories based on the settlement patterns. We also compare the footprint-derived classification to the 2016 Multidimensional Open Data Urban Morphology (MODUM) dataset [46]. MODUM was produced using self-organising maps to cluster multiple OA characteristics including building footprint measure summaries (density, adjacency, etc.) as well as an area's spatial relationship to land cover types and infrastructure such as railway stations or major roads.

## Results

The building footprint dataset contains over 13.8 million features. The processing steps with the *foot* package created forty-six separate gridded layers for each metric at a 100 m x 100 m spatial resolution in GeoTiff format (23 layers using the focal radius and 23 layers without). In addition to the summary measures, there is also a binary layer indicating pixels with one or more building footprints present and a raster with the counts of footprints present. An example of one of the gridded layers for the count of buildings within the 250 m focal window is shown in Fig 4. All the resulting output data are provided in the University of Southampton data repository at https://doi.org/10.5258/SOTON/D1674, and the script showing the *foot* commands is provided in the supplemental materials (S1 Code).

Fig 5 presents the result of the cluster analysis for three example locations. A full map and the classified raster dataset are included in the supplemental materials (S1 File). Minimising the BIC suggested 6 groups as the optimal number of clusters in the data. The labels on the classes are assigned randomly by the algorithm. This analysis highlights regions with similar morphology and patterns in the building footprints. Despite not using any explicitly spatial information on the location of the observations in the clustering approach, clear geographic patterns emerge during visual inspection of the results. Major urban areas, such as London and Manchester, emerge in class 4 with the core areas in class 5, though these classes are also found in some outlying city areas suggesting some similarity in settlement patterns. Types 3 and 6 appear to highlight the fringe of urban agglomerations, which could be useful for highlighting areas of potential growth or landscape change. Types 1 and 2 appear predominantly in sparsely settled and rural areas.

In order to compare the footprint-derived classes with existing classification systems, we aggregated the grids to the 2011 Census Output Areas by their majority type and created contingency tables with the RUC and MODUM datasets. Table 2 shows the percent of output

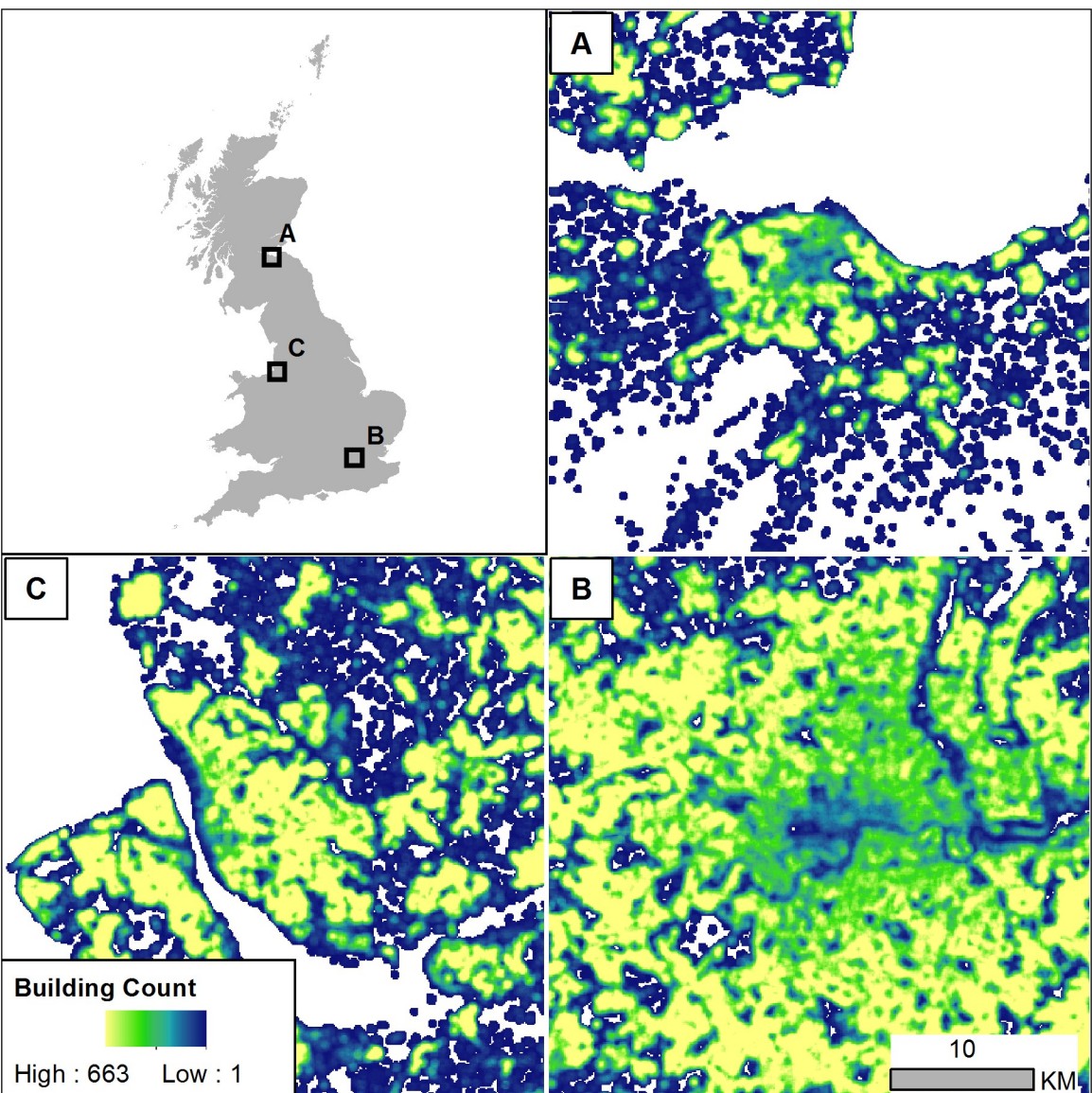

**Fig 4. Overview of gridded count of buildings calculated within a 250 m focal window using the R package *foot*.** Results shown are 100 m x 100 m spatial resolution gridded data. Examples of the results are shown for areas around Edinburgh (A), London (B), and Liverpool (C). Full datasets are provided in the supplemental materials. Data shown are the authors' calculations using building footprints and boundaries released by Ordnance Survey under the Open Government Licence (OGL) v3.0 (Contains OS data © Crown copyright and database right 2018, 2020).

areas in each of the RUC groups. Overall the footprint-derived classes seem to capture some of the urban-rural transition and gradient of different settlement types defined from the RUC data. Overall, the majority of OAs are split among classes 3, 4, and 6 with a much lower percentage of OA units classified as 1, 2, or 5 based on footprint-derived classes. In major conurbations defined by RUC, class 4 is most represented, in 41% of OAs. This settlement pattern type becomes gradually less common and is largely unobserved in isolated and sparse rural areas. In contrast, classes 1 and 2 gradually increase in prevalence in rural areas and seem to identify villages, hamlets, and sparse settings consistent with RUC data. Other notable patterns

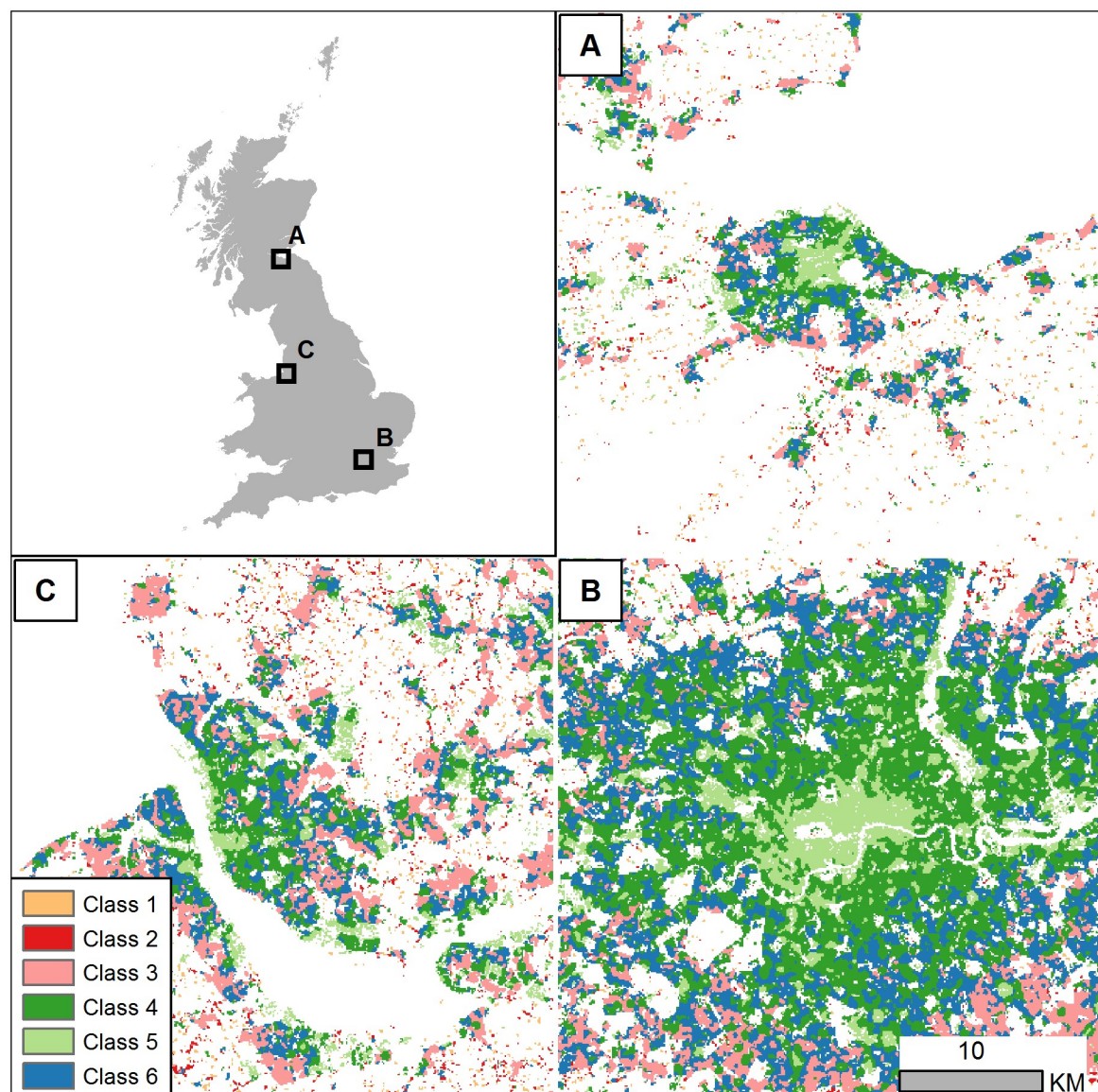

**Fig 5. Example of gridded settlement patterns.** The settlement types were created using unsupervised clustering methods to identify potential typologies based on morphology measurements of building footprint polygons. Full dataset provided in the supplemental materials. Data shown are the authors' calculations using building footprints and boundaries released by Ordnance Survey under the Open Government Licence (OGL) v3.0 (Contains OS data © Crown copyright and database right 2018, 2020).

include class 5 which is predominantly found in urban OAs, and as noted previously, is mainly observed as the core of these urban regions.

Table 3 shows the second Output Area comparison using the same footprint derived classes but compared the eight MODUM classes [46]. The first RUC comparison (Table 2) highlights the urban gradient and the transition to rural areas, while the MODUM clusters highlight more descriptive, functional areas within cities. Overall, the unsupervised footprint pattern-derived classes are less able to differentiate among the MODUM clusters, although the footprint classes do appear to differentiate between urban, suburban, and countryside which is consistent with the RUC comparison. For instance, while class 4 identifies central business district OAs in MODUM, it is also prevalent in highstreet and "railway buzz" areas suggesting

**Table 2. Comparison of footprint pattern classes and the 2011 census rural-urban classification for output areas in England and Wales.**

| | Rural Urban Classification of Output Areas (2011) | | | | | | | | | | | |
| | Urban | | | | Rural | | | | | | | |
| | Major conurbation | Minor conurbation | City and town | City and town in a sparse setting | Rural town and fringe | Town and fringe in a sparse setting | Village | Village in a sparse setting | Hamlets and isolated dwellings | Hamlets and isolated dwellings in a sparse setting | | |
| Footprint Classes | A1 (%) | B1 (%) | C1 (%) | C2 (%) | D1 (%) | D2 (%) | E1 (%) | E2 (%) | F1 (%) | F2 (%) | Sum | (%) |
|---|---|---|---|---|---|---|---|---|---|---|---|---|
| 1 | 0.1 | 0.2 | 0.4 | 1.6 | 2.1 | 2.2 | 19.8 | 24.1 | 46.1 | 63.4 | 6248 | 3.4 |
| 2 | 0.2 | 0.1 | 0.8 | 1.6 | 2.9 | 3.5 | 25.6 | 34.3 | 43.0 | 34.0 | 6967 | 3.8 |
| 3 | 20.1 | 42.2 | 37.8 | 31.8 | 57.0 | 36.0 | 32.8 | 21.2 | 4.0 | 0.7 | 58366 | 32.2 |
| 4 | 41.0 | 23.9 | 27.6 | 33.5 | 14.3 | 24.2 | 8.4 | 6.3 | 3.5 | 0.7 | 51917 | 28.6 |
| 5 | 7.7 | 3.2 | 3.4 | 4.7 | 1.0 | 0.6 | 1.5 | 0.4 | 1.3 | 0.1 | 7953 | 4.4 |
| 6 | 30.9 | 30.3 | 30.0 | 26.7 | 22.6 | 33.4 | 11.8 | 13.9 | 2.0 | 1.1 | 49948 | 27.5 |
| Sum (%) | 100 | 100 | 100 | 100 | 100 | 100 | 100 | 100 | 100 | 100 | 181399 | 100 |

Values are the percent of the Output Areas in each category. The gridded footprint classes were summarised to the Output Area polygons based on the majority type.

**Table 3. Comparison of footprint pattern classes and MODUM clusters for output areas in England Wales.**

| | MODUM Clusters (2016) | | | | | | | | | |
| | Central Business District | Countryside Sceneries | High Street and Promenades | Railway Buzz | Suburban Landscapes | The Old Town | Victorian Terraces | Waterside Settings | | |
| Footprint classes | (%) | (%) | (%) | (%) | (%) | (%) | (%) | (%) | Sum | (%) |
|---|---|---|---|---|---|---|---|---|---|---|
| 1 | 0.0 | 13.2 | 0.4 | 0.1 | 0.2 | 0.0 | 0.2 | 2.0 | 6248 | 3.4 |
| 2 | 0.0 | 13.9 | 1.0 | 0.4 | 0.5 | 0.0 | 0.2 | 3.4 | 6967 | 3.8 |
| 3 | 2.3 | 45.7 | 18.3 | 18.9 | 50.4 | 3.8 | 15.7 | 31.6 | 58366 | 32.2 |
| 4 | 64.1 | 11.2 | 40.2 | 42.7 | 17.1 | 51.8 | 40.5 | 27.9 | 51917 | 28.6 |
| 5 | 14.4 | 2.6 | 14.7 | 6.8 | 0.8 | 25.5 | 2.5 | 4.2 | 7953 | 4.4 |
| 6 | 19.2 | 13.5 | 25.4 | 31.1 | 31.0 | 18.9 | 40.9 | 31.0 | 49948 | 27.5 |
| Sum (%) | 100 | 100 | 100 | 100 | 100 | 100 | 100 | 100 | 181399 | 100 |

Values are the percent of the Output Areas in each category. The gridded footprint classes were summarised to the Output Area polygons based on the majority type.

that the footprint class is primarily highlighting a more general urban commercial district. Class 3, seen in Fig 5 (and S1 File) on the periphery of urban areas is the majority class in MODUM's definition of suburban landscapes.

## Discussion

This present work introduced *foot*, a new R package designed to help researchers to extract new information on settlement patterns and summarise building footprint datasets. The *foot* package provides building block functions for consistently calculating morphology measures at the building-level and summary measures in user-defined zones. It also provides a convenient set of functions to support common workflows. A basic set of morphology characteristics are currently available, and the package is open-source and it can incorporate additional

measures in the future. Development is ongoing and additional features of the package can be found in the online documentation: https://wpgp.github.io/foot/. At a minimum, a user of the package must be somewhat familiar with R, but our package removes the need for detailed programming and minimal spatial data management. More advanced users may find it useful to integrate the provided functions into their specialised models or workflows.

In particular, *foot* can create gridded representations of building morphology measures which have not been widely supported previously. As noted by Heris, Foks [39], utilising large vector databases of building shapes is computationally challenging for many applications. Grids, or raster datasets, provide a simplified representation of building datasets and are functionally similar to remote sensing data. Conceptually, gridded measures can help to develop a landscape perspective of the built environment with gradients in settlement types and patterns rather than in bounded, arbitrary units. From an analytical perspective, grids have advantages for more easily integrating with other data sets for spatial models.

To our knowledge, there are presently no other R packages to support building morphology calculations. The R language is growing in popularity and the *foot* package makes morphological calculations accessible to more users. It also avoids an alternative to complicated GIS work-flows in (potentially proprietary) GIS software. In the Python programming language the *momepy* package provides another toolkit for urban form analysis [47]. However, *momepy* is primarily designed for summary calculations within morphological tessellations [40] or similar areas such as cadastral plots and does not currently support gridded output datasets. Heris, Foks [39] provide gridded output layers calculated from the Microsoft building footprints for the United States. However, their summary metrics are limited to the total coverage, counts of buildings, the average, minimum and maximum areas in each grid cell. While they provide Python scripts to carry out their specific processing, they do not provide a package of general purpose tools.

We demonstrated the use of the *foot* package by efficiently calculating a variety of gridded morphology metrics using all building footprints from Great Britain. We then used the gridded layers in a simple analysis to derive a settlement typology, and we compared out prediction to census-derived and machine learning settlement classifications [46] for England and Wales. Deriving building uses [48], settlement types or neighbourhoods [49, 50] is one area of application using morphometric patterns. We used a Gaussian mixture model (GMM) as an unsupervised classification method to explore the patterns in the building morphology grids. Our analyses suggested six groupings best clustered the data. The results of the comparison between our six classes from the unsupervised clustering with census rural-urban classification do suggest that morphological patterns of building footprints can help to differentiate the urban gradient in this study region (from sparse settings to the urban core). The prevalence of the footprint-derived settlement classes varied across the rural-urban classification. For instance, two classes clearly identified hamlets and other sparse rural areas and were largely absent from urban areas. Conversely, major conurbations and cities and towns were predominately made up of three of the footprint-derived classes. There were also distinct visual patterns to the footprint classes as core urban areas were distinct from urban fringes. The similarities of our footprint-derived classifications to the 2011 RUC data is less surprising given that physical form is the focus of both datasets. Expanding the focal window used to calculate the building footprint metrics beyond 250 m, or using layers calculated at multiple buffer distances, may improve the classification further by identifying more of the broader context of sparse settings (e.g. small clusters of structures in close proximity to relatively larger settlements). Regardless, the result is promising for developing settlement maps based on footprint data in settings which lack detailed urban planning or census data. However, the comparison of our footprint derived classes with MODUM data suggests that, on their own,

the simple morphology metrics used are insufficient to differentiate certain intra-urban areas, such as activity centres associated with railway access. In our comparison, the footprint-derived classes differentiated rural-urban differences again, but showed considerable overlap with the intra-urban MODUM classes. For example, class 4 of the footprint patterns was the predominant type in MODUM's central business districts, high streets, railway buzz, and Victorian terrace classes suggesting that it is representing a more general "urban" pattern. The MODUM classification uses additional data on points of interest, amenities, road networks and population to derive eight classes. This approach provides a fuller picture of the local area, but it has higher data requirements which might limit where the approach can be applied. Our case study was primarily a demonstration of programming tools and not meant to replace more focused analyses such as MODUM, but it is important to highlight the potential limitations of using morphology-derived settlement classes.

Future research should continue to explore the key metrics, spatial scales, and representations needed to accurately identify key patterns in urban form for different contexts. The functionality of the *foot* package can support such work. We noted above that the *foot* package is continuing development and will be expanded to include additional metrics. We have limited our discussion to two-dimensional representations of buildings. These are the most common; however, 3D shapes are becoming available from LIDAR and radar data [51, 52]. Morphology measures may soon need to account for volumetric differences.

## Conclusion

Building footprint datasets are becoming more widely and openly available, covering entire countries. These data can be particularly valuable in low- and middle-income country settings which are experiencing rapid urban growth and change [53] and may lack other information on urban areas. Spatially-detailed polygon representations of structures are proving to be useful in a range of disciplines and applications; however, applied researchers and practitioners can benefit from having new tools such as the R package *foot* to effectively and efficiently work with building footprint datasets and extract morphological information.

## Supporting information

**S1 Text. Installation instructions.**
(DOCX)

**S2 Text. Vignettes and documentation from *foot*.**
(PDF)

**S1 Code. Processing script for the case study.**
(R)

**S1 File. Case study settlement type layer as a PDF map.**
(PDF)

## Acknowledgments

The authors acknowledge the use of the IRIDIS High Performance Computing Facility, and associated support services at the University of Southampton, in the completion of this work. The authors thank Edith Darin, Claire Dooley, Attila Lazar, and Douglas Leasure for reviewing earlier versions of the R package.

## Author Contributions

**Conceptualization:** Warren C. Jochem, Andrew J. Tatem.

**Formal analysis:** Warren C. Jochem.

**Funding acquisition:** Andrew J. Tatem.

**Methodology:** Warren C. Jochem, Andrew J. Tatem.

**Software:** Warren C. Jochem.

**Supervision:** Andrew J. Tatem.

**Writing – original draft:** Warren C. Jochem, Andrew J. Tatem.

**Writing – review & editing:** Warren C. Jochem, Andrew J. Tatem.

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
