## [Decision Letter · Decision Letter 0]

18 Jan 2021

PONE-D-20-38873

Tools for mapping multi-scale settlement patterns of building footprints: An introduction to the R package foot

PLOS ONE

Dear Dr. Jochem,

Thank you for submitting your manuscript to PLOS ONE. After careful consideration, we feel that it has merit but does not fully meet PLOS ONE’s publication criteria as it currently stands. Therefore, we invite you to submit a revised version of the manuscript that addresses the points raised during the review process.

The reviewers see merit in the manuscript, but suggest some revisions. Please make those changes to the manuscript.

We look forward to receiving your revised manuscript.

Kind regards,

Wenhao Yu, Ph.D.

Academic Editor

PLOS ONE

Additional Editor Comments:

The reviewers see merit in the manuscript, but suggest some revisions. Please make those changes to the manuscript.

Journal Requirements:

3. We note that Figures 1-5, Supplementary File 2 (F1, F2), Supplementary File 4 in your submission contain [map/satellite] images which may be copyrighted. All PLOS content is published under the Creative Commons Attribution License (CC BY 4.0), which means that the manuscript, images, and Supporting Information files will be freely available online, and any third party is permitted to access, download, copy, distribute, and use these materials in any way, even commercially, with proper attribution. For these reasons, we cannot publish previously copyrighted maps or satellite images created using proprietary data, such as Google software (Google Maps, Street View, and Earth). For more information, see our copyright guidelines: http://journals.plos.org/plosone/s/licenses-and-copyright.

3.1.    You may seek permission from the original copyright holder of Figures 1-5, Supplementary File 2 (F1, F2), Supplementary File 4 to publish the content specifically under the CC BY 4.0 license. 

3.2.    If you are unable to obtain permission from the original copyright holder to publish these figures under the CC BY 4.0 license or if the copyright holder’s requirements are incompatible with the CC BY 4.0 license, please either i) remove the figure or ii) supply a replacement figure that complies with the CC BY 4.0 license. Please check copyright information on all replacement figures and update the figure caption with source information. If applicable, please specify in the figure caption text when a figure is similar but not identical to the original image and is therefore for illustrative purposes only.

Reviewers' comments:

Reviewer's Responses to Questions

**Comments to the Author**

1. Is the manuscript technically sound, and do the data support the conclusions?

Reviewer #1: Yes

Reviewer #2: Yes

2. Has the statistical analysis been performed appropriately and rigorously? 

Reviewer #1: N/A

Reviewer #2: Yes

3. Have the authors made all data underlying the findings in their manuscript fully available?

Reviewer #1: Yes

Reviewer #2: Yes

4. Is the manuscript presented in an intelligible fashion and written in standard English?

Reviewer #1: Yes

Reviewer #2: Yes

5. Review Comments to the Author

Reviewer #1: This is an interesting approach to mapping, and I believe it will achieve positive results in locations where large-scale planning has not been implemented. Countries with limited data and mapping capabilities will benefit from this research.

Reviewer #2: Overall the manuscript is well-written and sufficiently descriptive. The R package described appears to be a useful contribution for a variety of research and applications. I only have a few requests before recommending for publication:

- Please provide a description of morphology parameters that are not obvious. Specifically shape and compactness. Even if these are exhaustively described in cited literature, a brief description should be included here.

- Line 153 - provide an equation for how entropy is calculated in this work

- Expand the conclusions to summarize findings from comparison of the clustering effort demonstrated in this paper with the Census and MODUM data described in the paper.

- Line 254 - what is meant by proxy datasets?

- Line 239 - change "be" to "been"

6. PLOS authors have the option to publish the peer review history of their article (what does this mean?). If published, this will include your full peer review and any attached files.

Reviewer #1: No

Reviewer #2: No

---

## [Author Response · Author response to Decision Letter 0]

2 Feb 2021

Response to Reviewers

Dear Dr. Yu and Reviewers,

Thank you for your time and attention to reviewing our manuscript. We appreciate your comments and questions and that you have found merit in our work. Below we are addressing each concern in a point-by-point response. From here, our responses to comments will be marked in red font. Thank you for your consideration. 

Yours sincerely,

Warren Jochem

Additional Editor Comments:

We have reviewed the style requirements and reformatted our manuscript and our file names, especially the additional figures and figure captions. 

As requested in your email, we have submitted our figures for verification through the PACE system prior to uploading them to this revision.

We understand this requirement and will provide the necessary DOI numbers to the Editor on acceptance of our manuscript. Our files have already been deposited in a repository through our institution, which will grant the numbers once the manuscript has been accepted. Temporary/placeholder numbers have currently been assigned and our institution is waiting for the manuscript to be accepted.

3. We note that Figures 1-5, Supplementary File 2 (F1, F2), Supplementary File 4 in your submission contain [map/satellite] images which may be copyrighted. All PLOS content is published under the Creative Commons Attribution License (CC BY 4.0), which means that the manuscript, images, and Supporting Information files will be freely available online, and any third party is permitted to access, download, copy, distribute, and use these materials in any way, even commercially, with proper attribution. For these reasons, we cannot publish previously copyrighted maps or satellite images created using proprietary data, such as Google software (Google Maps, Street View, and Earth). For more information, see our copyright guidelines: http://journals.plos.org/plosone/s/licenses-and-copyright.

3.1. You may seek permission from the original copyright holder of Figures 1-5, Supplementary File 2 (F1, F2), Supplementary File 4 to publish the content specifically under the CC BY 4.0 license. We recommend that you contact the original copyright holder with the Content Permission Form (http://journals.plos.org/plosone/s/file?id=7c09/content-permission-form.pdf) and the following text:

3.2. If you are unable to obtain permission from the original copyright holder to publish these figures under the CC BY 4.0 license or if the copyright holder’s requirements are incompatible with the CC BY 4.0 license, please either i) remove the figure or ii) supply a replacement figure that complies with the CC BY 4.0 license. Please check copyright information on all replacement figures and update the figure caption with source information. If applicable, please specify in the figure caption text when a figure is similar but not identical to the original image and is therefore for illustrative purposes only.

Thank you for directing us to the policies and resources regarding licensing of images and map data in PLOS One. This manuscript uses only open data, licensed with the Open Government License version 3.0 (http://www.nationalarchives.gov.uk/doc/open-government-licence/version/3/) and the Open Database License (https://opendatacommons.org/licenses/odbl/ ). These licenses are, to our understanding, compatible with CC-BY, allowing copying, re-distribution, adaptation, and re-use (including commercial uses). They are attribution-only licenses which meets the requirement stated in section 3.2 above. 

We have amended the manuscript to make the licensing and data sources clear in several places. First, in the main text (lines 137-138, lines 322-323) we have added a note on the source of the data when describing the examples and case study. Second, in each figure caption using these data, we have added text giving the specific data source, copyright and license information.

Reviewers' comments:

Reviewer #1: This is an interesting approach to mapping, and I believe it will achieve positive results in locations where large-scale planning has not been implemented. Countries with limited data and mapping capabilities will benefit from this research.

We thank the reviewer for their time in reviewing our manuscript.

Reviewer #2: Overall the manuscript is well-written and sufficiently descriptive. The R package described appears to be a useful contribution for a variety of research and applications. I only have a few requests before recommending for publication:

We thank the reviewer for their helpful comments and suggestions.

- Please provide a description of morphology parameters that are not obvious. Specifically shape and compactness. Even if these are exhaustively described in cited literature, a brief description should be included here.

We have amended the manuscript to include a more details of the morphology metrics we use. On lines 156 – 162 we have added text describing the shape and compactness measures as well as the nearest neighbour index in lines 180 – 181. These additions include the questions, describing the potential range of values as well as added references to where the measures were developed. As noted below, we have also expanded our explanation of the entropy measure.

- Line 153 - provide an equation for how entropy is calculated in this work

In addition to providing a fuller explanation for the morphology metrics discussed above, we have added the specific equations for entropy and other calculations implemented in our R package (lines 171 – 173). For this work we use Shannon entropy.

- Expand the conclusions to summarize findings from comparison of the clustering effort demonstrated in this paper with the Census and MODUM data described in the paper.

Thank you for this suggestion. We have expanded our discussion section to summarise the findings of our comparisons and to provide more context to the interpretation of our results. Specifically we note that our footprint-derived clusters are consistent with the urban-rural classification gradients and less similar to the MODUM data. This finding is not surprising given that features related to physical form are important in the census classification. However, MODUM draws on additional data sources, and it’s important to consider that such requirements may limit its wider application. A simpler typology deriving solely from footprint patterns (as we demonstrated) could be implemented in more areas. We incorporated these edits into the original discussion section, but the additions are primarily in lines 476 – 514 of the revised manuscript.

- Line 254 - what is meant by proxy datasets?

Thank you for raising this question. We were referring to something like a “file pointer” as the program only requires the location of the input footprint data and does not have to read the entire contents of large files. We agree that wording was unclear and have changed the manuscript. In line 284 of the revised manuscript we have removed the phrase “proxy datasets” as we feel the sentence still conveys the important information that only sub-datasets are extracted and analysed, potentially in parallel. 

- Line 239 - change "be" to "been"

We have made this change and reviewed the manuscript again for any other grammar and spelling errors. Thank you.

---

## [Editor Report · Decision Letter 1]

9 Feb 2021

Tools for mapping multi-scale settlement patterns of building footprints: An introduction to the R package foot

PONE-D-20-38873R1

Dear Dr. Jochem,

We’re pleased to inform you that your manuscript has been judged scientifically suitable for publication and will be formally accepted for publication once it meets all outstanding technical requirements.

Kind regards,

Wenhao Yu, Ph.D.

Academic Editor

PLOS ONE

Additional Editor Comments (optional):

Thanks for your revision.
---

## [Editor Report · Acceptance letter]

17 Feb 2021

PONE-D-20-38873R1 

Tools for mapping multi-scale settlement patterns of building footprints: An introduction to the R package *foot*

Dear Dr. Jochem:

I'm pleased to inform you that your manuscript has been deemed suitable for publication in PLOS ONE. Congratulations! Your manuscript is now with our production department. 

Kind regards, 

on behalf of

Dr. Wenhao Yu 

Academic Editor

PLOS ONE